# Global Health Challenges: Why the Four S’s Are Not Enough

**DOI:** 10.3390/children9121867

**Published:** 2022-11-30

**Authors:** Nathan M. Novotny, Shadi Hamouri, Donna Kayal, Sadi Abukhalaf, Haitham Aqra, Wael Amro, Ahmad Shaltaf

**Affiliations:** 1Beaumont Children’s, Section of Pediatric Surgery, Oakland University William Beaumont School of Medicine, Royal Oak, MI 48073, USA; 2Department of General Surgery and Urology, Faculty of Medicine, Jordan University of Science and Technology, Kind Abdullah University Hospital, Irbid 22110, Jordan; 3William Beaumont School of Medicine, Oakland University, Rochester, MI 48309, USA; 4Section of Pediatric Surgery and Anesthesiology, Palestine Medical Complex, Ramallah P601, Palestine; 5Department of General Surgery and Special Surgery, Faculty of Medicine, Al-Balqa’ Applied University, Al-Salt 19117, Jordan; 6Department of Surgery, University of Illinois Metropolitan Group, Chicago, IL 60607, USA

**Keywords:** thoracoscopy, global health, complications, minimally invasive surgery

## Abstract

A well-known tenant of global health is the need for the four-S’s to be successful in providing care in any context; Staff, Stuff, Space and Systems. Advanced thoracoscopy is slow to gain traction in low- and middle-income countries (LMICs). To our knowledge, no pediatric advanced thoracoscopy had been attempted previously in either LMIC. Therefore, we report the challenges associated with the adoption of the first advanced thoracoscopic procedures in two LMIC hospitals by a visiting surgeon. To further identify aspects of care in promoting the introduction of advanced thoracoscopy, we added a fifth S as an additional category—Socialization. A key to accomplishing goals for the patients as a visiting surgeon, particularly when introducing an advanced procedure, is acceptance into the culture of a hospital. Despite facing significant obstacles in caring for complex thoracic pathology with heavy reliance on disposable and reusable instrumentation provided through donation and limitations in staff such as access to neonatologists and pediatric surgeons, many obstacles have been overcome. In this perspective article, we show that a “fifth S” is also integral—having local surgeons and anesthesiologists eager to learn with acceptance of the visiting surgeon’s expertise opens a path towards attempting advanced procedures in limited-resource settings.

## 1. The Four S’s of the Global Health Tenant

A well-known tenant of global health is the need for the so called “four-S’s” to be successful in providing care in any context; (1) Staff (number of physicians, nurses and specialists qualified to care for the patient), (2) Stuff (available resources), (3) Space (capacity given patient volume) (4) Systems (well-structured and standardized system for triage and resource allocation) [1]. These four categories are applicable to any care given, and the structure is helpful in assessing preparedness of healthcare facilities during disaster-responses, pandemics, such as the recently encountered COVID-19 pandemic, or implementing new forms of advanced procedures [1].

Minimally invasive surgery (MIS) is a standard of care for many pediatric surgical conditions in high-income countries (HIC) [2]. Rates of adoption for many common procedures are increasing and have shown advantages for many pediatric surgical conditions [2,3,4,5]. MIS is associated with reduced surgical site infection rates, reduced post-operative pain, a decrease in hospital stay and ultimately earlier return to daily activities which are advantageous particularly when considering low- and middle-income countries (LMICs) [6,7]. Because of this, many LMIC surgeons are beginning to adopt minimally invasive techniques for their patients [8,9,10,11]. Though, it is important to note that care must be taken to introduce these challenging procedures in an ethical, thoughtful, sustainable, and stepwise fashion with proper mentoring [12,13,14,15].

Thoracoscopy is arguably more challenging than laparoscopy for both the surgeon and the anesthesiologist for a myriad of reasons. The alterations in physiology caused by the pneumothorax, the less familiar anatomy and the impossibility of externalizing most of the thoracic structures, unlike the abdominal organs, are just a few of the challenges posed by thoracoscopic procedures [14,16]. The varying ages of patients cared for by pediatric surgeons also provides a challenge to surgeon and anesthesiologist alike [16]. Instruments of reduced size are often indicated for patients less than 2 years of age, ventilation techniques differ based on the child’s functional residual capacity given their age, and airway management are a few of the characteristics taken into consideration [17,18]. Additionally, post-operative care can be more challenging for care providers unfamiliar with chest tubes and more acutely-presenting complications such as tension pneumothorax.

One HIC pediatric surgeon who is experienced and facile in advanced thoracoscopy worked as a ‘surgeon in residence’ in two LMIC hospital settings in the Middle East and attempted to introduce and teach advanced thoracoscopy in each. In an effort to not repeat the infamous mistakes during the initial adoption of laparoscopic cholecystectomy in the United States, it is incumbent on those with existing skills (often but not limited to the HICs), to help those who have little or no experience to adopt these new techniques prudently to provide the benefit of MIS to even the first patients receiving the care [19]. In this process, the four global health tenants become increasingly critical in implementing advancements through the experience of the HIC surgeon. There were many opportunities for improvement and the experiences in each of the two hospitals are recounted with an emphasis on identifying aspects of care that promoted the introduction of advanced techniques, facets that could be improved and difficulties that could be prevented in future attempts.

## 2. First Hand Experience in a Global Health Initiative: The Two LMICs

To provide an insight into the importance of the “Four S’s” when implementing a new procedure, an IRB-approved retrospective chart review of a prospectively recorded database of the advanced thoracoscopic procedures performed in the hospitals of two middle-income countries in the Middle East between January 2017 and October 2018 was conducted. Hospital A (King Abdullah University Hospital) is a large hybrid public/private tertiary care center located in Irbid, Jordan, with many specialists trained in Western countries. Hospital A is the largest medical center in northern Jordan, serving approximately one million inhabitants from numerous cities with 20 Nursery beds, 10 NICU (Neonatal Intensive Care Unit) beds and 10 PICU (Pediatric Intensive Care Unit) beds [20]. Hospital B (Palestine Medical Complex) is a medium sized public tertiary center located in Ramallah, Palestine, with specialists trained in a variety of middle-income countries. Hospital B consists of five hospitals joined together serving the surrounding population of the city, with 15 NICU beds, 8 PICU beds and 10 surgical ward beds [21].

Patient selection in both hospitals was determined by the local pediatric surgeons based on presentation of patients necessitating the procedures during the presence of the surgeon in residence. Pediatric surgeons in Hospital A have intermediate level of experience performing laparoscopic surgery for many years and pediatric surgeons in Hospital B had low-intermediate level of experience performing supervised laparoscopic surgery for a few years, with limited to no experience in complex cases in both hospitals. The visiting surgeon in residence is a U.S. board certified pediatric surgeon skilled in advanced MIS. Patient demographics, operative characteristics, postoperative occurrences, and the postoperative debriefing of each case were thoroughly reviewed.

Over the 22-month period during which the visiting pediatric surgeon was in attendance at the two hospitals, four patients underwent five attempts at complex thoracoscopic procedures. In Hospital A, one TEF patient was referred to the visiting surgeon. There is unclarity on how many TEFs are performed in the hospital, as Hospital A is a government hospital with a private sector that would take the more affluent patients that can pay out of pocket. Additionally, Hospital A is not the only hospital that was performing TEF procedures and there are other surrounding hospitals that performed advanced thoracic procedures as well, although not thoracoscopically. Hospital B had multiple referrals to the visiting surgeon. In total, three thoracoscopic type C TEF repairs were attempted in newborns. Two were completed thoracoscopically and one was converted to open. One thoracoscopic aortopexy was performed in a four-week-old and one thoracoscopic right lower lobectomy was performed in a three-year-old. Table 1 summarizes the patient demographics and Table 2 summarizes the clinical and perioperative characteristics of the patients that underwent the thoracoscopic procedures during this period.

## 3. Patient Outcomes

Patient 1, Hospital A: Attempts were made to discuss the case prior to the operation with the anesthesiologist on call, however, the discussion never occurred. During the procedure, consistent visualization of the pertinent anatomy was unattainable given the method of ventilation. After 45 min, the azygous was divided and the fistula was clipped, however dissection of the upper pouch was unfeasible due to the lack of visualization in spite of intermittent apnea. Hence, the decision to convert to an open procedure was indicated. He was extubated on postoperative day 13 and had stridor, tachypnea and inability to wean from high flow oxygen. At four weeks of age, rigid bronchoscopy revealed severe tracheomalacia. A successful thoracoscopic aortopexy was performed in collaboration with the local adult thoracic surgeon. Patient 1 was rapidly weaned from oxygen and his stridor completely resolved.Patient 2, Hospital B: Prior to the operation, a conference was held with a HIC anesthesiologist experienced with thoracoscopic TEF anesthetics, as this patient was the first thoracoscopic TEF performed in the country. During the procedure, visualization was never obscured due to the anesthetic and the case was completed at an appropriate pace. On postoperative day 7, Patient 2 was given water soluble contrast at the bedside and a single portable chest x-ray was obtained as fluoroscopy was unavailable. No leak was identified and the patient’s diet was advanced, leading to discharge on postoperative day 10. At 4 weeks of age, Patient 2 returned with respiratory distress and a large right pleural effusion. Attempts to access the pocket of fluid percutaneously and thread the wire with ultrasound guidance was unsuccessful with the lack of an interventional radiologist. Therefore, it was decided to take the patient back to the OR and perform thoracoscopy to provide more control during tube placement. This was accomplished, however the fluid was mixed with blood from the difficult thoracoscopy which made the character and amount difficult to ascertain. The tube was then discontinued after two days of bloody drainage subsided. The patient was discharged on hospital day 4 and returned to clinic for a follow up two weeks later tolerating breast milk.Patient 3, Hospital B: This was the second thoracoscopic TEF performed in the country. The same anesthesiologist and anesthetic plan from Patient 2 was utilized given the previous successful procedure. During the procedure, the local pediatric surgeon was able to ligate the azygous vein, clip the fistula and start the dissection of the upper pouch without the assistance from the visiting pediatric surgeon. The case was then completed by the visiting surgeon and the patient was taken to the NICU postoperatively. On postoperative day 3, the patient desaturated for unclear reasons and was bagged in the NICU by the pediatrics resident but soon after arrested. A chest X-ray at the time showed a large left sided pneumothorax with a stable right sided chest tube without change in effluent. The pediatrics team attempted placement of a left sided chest tube but the pneumothorax was not evacuated. At the arrival of the pediatric surgical team to the NICU, a larger left sided chest tube was placed for a pneumothorax but unfortunately, the team was unable to regain spontaneous circulation.Patient 4, Hospital B: Prior to the procedure, a preoperative conference was held with the anesthesiologist to outline the goals and operative plan for the right lower lobe CPAM. During the procedure, 3 mm instruments were used with an additional single 5 mm trochar to allow the use of a 5 mm vessel sealer and a 5 mm stapler for the bronchus and larger vessels. The operation was completed in 90 min and the patient was extubated in the OR. On postoperative day 3, Patient 4 was successfully discharged, with no further complications noted during the follow up visits.

### 3.1. The Four S’s Are Not Enough

Opportunities for improvement in each LMIC were categorized into one of each of the “four-S’s” (Staff, Stuff, Space, or Systems) and attempts were made to rectify the shortcoming prior to the next case. However, through the visiting surgeon’s first-hand experience, the four S’s were not enough for a HIC surgeon introducing advanced thoracoscopy in an LMIC and a critical component was recognized in the differences observed between the two LMICs.

### 3.2. Fifth S, Socialization

Multiple barriers exist to integration of a HIC surgeon into a LMIC hospital. However, a major aspect of care in promoting introduction of any procedure, in this case, advanced thoracoscopy, is Socialization (overcoming cultural and integration barriers). As seen in every aspect of healthcare, acceptance into the setting of a hospital in any region is critical in the success of an individual, but more importantly the overall organization. If experienced physicians and surgeons are to help effect change in LMICs, they must be accepted and trusted. Cultural barriers cannot be understated and are unique to each context, hence community engagement is key [22]. Global political pressures also vary greatly from one country to another and must be taken into each context of a visiting surgeon. Therefore, beyond the known “four-S’s” (Staff, Stuff, Space, and Systems), a fifth S is proposed—Socialization.

To address the importance of the global health tenants in both LMICs, an evaluation of each of the characteristics of the “Four-S’s” in addition to the proposed Fifth S, Socialization, was conducted. This was an essential element for evaluating the introduction of advanced thoracoscopy in LMICs. Table 3 summarizes the existing and missing characteristics for each category.

## 4. Challenges with Each Global Health Tenant

Significant challenges still exist in the care of neonates and children in the LMICs [23]. While outcomes cannot be expected to be similar to HICs, LMICs must continue to strive to implement quality improvement and adoption of proven techniques in caring for neonates and children. In the four cases presented, no complication experienced was clearly due to the surgical approach. Additionally, for each of these cases, they were the first known attempts in the history of the two countries. For four of the five procedures completed thoracoscopically, each was a technical success and evidence that advanced thoracoscopic procedures can be feasible in the LMICs context with proper preparation and consideration for the global health tenants.


Staff


Challenges specific to staffing include interest in local physicians and surgeons in collaboration and attempting a new approach. Hospital A’s adult thoracic surgeon showed interest in attempting newer techniques but the anesthesia and pediatric surgery departments were reluctant. Due to this, difficulties arose in adequately preparing all providers who contribute to the success or failure of the procedure. Despite the lack of proper preparation from anesthesia, the child’s safety was never compromised in the attempt at thoracoscopic repair. Post conversion, the neonate received a standard open TEF repair and suffered no adverse consequences from the attempt at thoracoscopic repair.

In Hospital B however, challenges with staffing differed in that the local pediatric surgeons’ experiences with minimally invasive procedures came entirely from multiple previous short-term visits by the visiting pediatric surgeon from the U.S. Because of this, there was limited participation by the local pediatric surgeons initially, however, that gradually increased as they expanded their experience with the procedures. The goal is for the local pediatric surgeons to be able to perform the full gamut of MIS independent of the visiting pediatric surgeon. Indeed, the ultimate goal is for the local pediatric surgeons to implement the first pediatric surgical fellowship to populate the area with fully trained pediatric surgeons facile in both open and minimally invasive surgeries.

Additionally, in Hospital B, anesthesiology was open and eager to understand the differences between thoracoscopic versus open surgeries. The anesthesiologists’ willingness to learn new methods of delivering a unique anesthetic provided the largest advantage to accomplishing these procedures at Hospital B. Furthermore, Hospital B distinguished itself from Hospital A in that the pediatric surgeons had a larger desire to learn from the visiting pediatric surgeon and pursued this opportunity. Prior to the visiting surgeon’s first trip, they had performed only two laparoscopic appendectomies, and after adequate mentoring, are currently performing mid-level laparoscopy and thoracoscopy independently.

The largest challenge encountered with staffing at Hospital B contributed to the series’ only mortality. The lack of neonatologist at the hospital was likely directly related to the series of events leading to this neonate’s demise. It is important to avoid placing the blame on the pediatric residents that staffed the NICU, however, the system failed the staff (and neonate), in this situation. The inexperienced house-staff were unable to adequately care for a neonate that was so urgently and critically ill. While it does seem that the left sided pneumothorax was due to barotrauma during attempts to increase the neonate’s oxygen saturation, having adequately trained staff could possibly have prevented this regrettable complication. Another potential for intervention could have come from the surgical team. However, the surgical team is also understaffed and was unable to respond in a timely manner, therefore the complete evacuation of the pneumothorax was severely delayed. Fortunately, the hospital has since recruited a neonatologist, and the team is optimistic that such a complication is less likely to occur in the future. However, recruitment to the government hospital system continues to be an ongoing problem due the wide disparity in pay between government hospitals and the private system.


Stuff


Particularly for procedural specialties, equipment can mean the difference between success and failure. Hospital A had adequate supplies to complete most advanced thoracoscopic procedures because of their ability to do neonatal laparoscopy. Hospital B on the other hand acquired their MIS equipment over four years of short-term trips by the visiting pediatric surgeon including some equipment purchased by the hospital directly. In addition, donations from hospital systems (Beaumont Health, Michigan), major instrument manufacturers (Karl Storz) and support from pediatric-specific instrument manufacturers (JustRight and Bolder Surgical) helped equip Hospital B to perform MIS. Without these donations, the hospital would have taken years to cobble enough minimally invasive equipment for basic MIS.

In equipping Hospital B, every effort was made to utilize reusable instruments to assure a more sustainable effort to decrease reliance on continued infusion of disposable instruments. However, pediatric-specific staplers and sealers are nearly indispensable for cases like thoracoscopic lobectomies. Furthermore, the lack of fluoroscopy at Hospital B continues to be a concern, nevertheless, the staff are trialing using the OR’s C arm for these studies.


Space


Having a space to provide the care is integral to providing the care [9]. Fortunately, space was not a consistent limiting factor in either hospital for performing advanced thoracoscopy with respect to operating, neonatology care, or pediatric intensive care. Hospital B however does have trouble with space for elective outpatient procedures which often result in a two-year waiting list for elective surgical procedures. This, however, did not contribute to delaying the care in urgent/emergent cases performed at the hospital.


Systems


Hospital A’s system for assigning an anesthesiologist to cases proved challenging during the two years the visiting pediatric surgeon was on site. Although an electronic medical record exists, a formal system for initiating consultation with anesthesiologist was not well established, especially for urgent cases.

Hospital B relies heavily on the pediatric surgeons for cleaning, sterilization, and storage of the pediatric-specific minimally invasive instruments. Additionally, equipping the hospital with disposable and non-disposable instruments depended on outside donations, which is not a sustainable model. Much work remains to convince an already financially limited government hospital to invest in (often) expensive minimally invasive instrumentation as reusable instruments break and age.

## 5. The Importance of the Fifth S, Socialization

The two hospitals are separated by less than 60 m but the political situation and perceived power of each people is tremendously different which affects their willingness to accept outside help. Hospital A’s sophistication also comes with a surety and independence that can pose a challenge when outside help is offered. Hence, only one TEF patient was referred to the visiting pediatric surgeon due to the reluctance from both the pediatric surgeons and anesthesiologists in accepting the visiting pediatric surgeon’s help.

Hospital B’s government arguably has a worse relationship with the government of the visiting pediatric surgeon, but their lack of formal training lends itself more readily to accepting help from outside. The local recognition of need is key. From the senior author’s experiences, the recognition of need in middle income countries is more heterogenous compared to a more uniform recognition of need and willingness to accept outside help seen in lower income countries. Without a recognition of need, the visiting surgeon faces a challenging situation when attempting to provide assistance.

The responsibility of a successful socialization however does not fall completely on the host country. The visiting surgeon must come respecting the host country’s people and culture so as to not offend the hosts [24]. The visiting surgeon should come with humility; equally a learner and a teacher given the specific context where they are practicing is foreign to them [24,25,26]. Additionally, prioritizing the host country’s interests above the surgeon’s own interests in all situations can only aid in helping develop a fragile initial development of a long-term relationship to build infrastructure in the host country via the “four-S’s” resulting in sustainable good. The better the visiting surgeon is socialized and accepted in the host hospital and culture, the better collaboration will be that will result in the most impactful change in the host institution as was directly experienced by the visiting pediatric surgeon.

## 6. The Importance of Promoting Advancements in LMICs: Why the Fifth S Is Needed

In the context of both Hospital A and B, it is crucial to understand that if a system is lacking the infrastructure, HIC surgeons presenting to LMICs in the context of humanitarian or teaching missions should be wary of introducing more advanced procedures that surpass the capabilities of the low-resource setting. However, in the setting in which the infrastructure and facilities are available, advancements can and should be made in expanding the capacity for provision of care [27]. The population of children in many LMICs is continuously expanding, but there is a significant unmet need for pediatric surgical services that is particularly visible in the context of congenital anomalies, acquired diseases and traumas [28,29]. Additionally, there is a significant deficit and unclarity in the workforce of pediatric surgeons in many LMICs, which prompts the necessity of introducing and expanding the interest in pediatric surgical training in such locations [27,29]. Within the context of the two hospitals, the infrastructure and workforce are available, but the paucity of advanced thoracoscopy was primarily due to lack of equipment due to costs and training [27,29]. As identified in numerous LMICs, the training of local surgeons may not fill the workforce gap faced by a given country, but builds a more effective system that allows for the advancement and development of LMICs that already have the necessary infrastructure [29]. Multiple studies have shown that humanitarian missions are not often effective in leading advancements, as the focus is on the number of cases performed rather than training of the local surgeons [27,28] However, as we presented through the visiting surgeon’s first-hand experience, the aim is to introduce and train the local surgeons in advanced thoracoscopic MIS given their experience in MIS. Several models have been introduced to improve the quality of surgical services in pediatric care, including teaching workshops, surgical camps, online training models and long-term partnership [27,28]. The various forms of such trainings have provided efficacious tools to leverage the skills of surgeons in LMICs, however, as faced in the two hospitals, the gains in expanding pediatric surgical capacity were also limited by the pediatric anesthesia. It is well known that surgical care goes hand in hand with anesthetic care, and any effort in expanding an LMIC’s surgical capacity must be matched with similar efforts in expanding anesthesia’s capacity to avoid anesthesia-related perioperative deaths [27,28]. However, as experienced with the visiting pediatric surgeon, socialization was a critical component in this context. The anesthesiologists had the skills necessary to successfully lead advanced MIS procedures, but their interest in expanding their training with the acceptance of an HIC surgeon who took on the role of mentoring is critical [27].

## 7. Ethical Dilemma in Global Health Initiatives

Arguably, the series’ only mortality in Hospital B brings up the ethical dilemma of introducing advanced MIS in the setting of an LMIC. The fine line between beneficence and non-maleficence is at the root of every surgeon’s duty [28,30,31]. A frequent concern in the expansion of surgeon capacity in LMICs is whether to leave a patient untreated due to the lack of infrastructure available to support their perioperative care [28]. However, in the hospital, the infrastructure was in place, but the system failed. Surgical procedures and advancements must be weighed in every context, and in the setting of the neonates in Hospital B, there was no other option. The visiting surgeon faced the dilemma of accepting death given the anomalies in consideration and the immediate requirement for surgical repair, or providing the tools and training to expand the capacity of the local surgeons in treating such anomalies [32]. Complications leading to mortality is an inevitable risk in every surgical procedure performed, however, that should not be a primary limitation in expanding the abilities and surgical skills in an LMIC. Additionally, the political environment surrounding the population in Hospital B is of critical consideration. Cost is a major prohibitive factor for many patients as private hospitals are not directly covered by governmental insurance, however, the option to transfer is also a limitation [33,34]. Access to healthcare for children is constrained due to the requirement for special permits and visas to travel to specialist hospitals, adding another burden for many children and their families given their restricted mobility [34,35]. Given the necessity for immediate surgery given the anomalies presented, such a transfer becomes impossible and disrupts effective health system functioning, which adds to the necessity of expanding the abilities of the LMIC pediatric surgeons and residents. With such considerations, it becomes critical in the face of every surgeon practicing in an environment with constrained resources to take into account the context in their evaluation of introducing surgical procedures. In Hospital B’s setting, the socialization aspect played a significant role in leading efforts to introduce advanced thoracoscopy with the local surgeons that can be sustainable. Because of the political situation of Hospital B, transfers to higher levels of care are challenging and expensive, sometimes prohibitively so. Because of this, it is common for open thoracic procedures to be performed in the imperfect system and with imperfect staffing. In stark contradistinction, the lack of socialization at Hospital A prevented the successful completion of one of the procedures and there was really no transfer of skill to the local pediatric surgeons in spite of the NICU’s better staffing and system to care for more critically ill babies.

## 8. Conclusions

The multifaceted nature of engaging in global surgery in a sustainable way is peppered with land mines. Beyond the simple technical challenges of doing complex procedures for the first time in a LMIC hospital, the barriers of Staff, Stuff, Space, Systems and Socialization each contribute to a sometimes seemingly unsurmountable task. Careful attention by a visiting surgeon to each of these areas can result in successful implementation of advanced procedures, in this case, advanced thoracoscopy with appropriate mentoring of locals from experienced surgeons and anesthesiologists. To sustain and continue promoting advanced MIS procedures by visiting HIC surgeons in LMICs, consideration to each of the global health tenants including socialization must be deeply accounted for. When each component is strongly built, successful implantation of advanced procedures become at reach.

## Figures and Tables

**Table 1 children-09-01867-t001:** Patient Demographics.

	PATIENT 1 *	PATIENT 2	PATIENT 3	PATIENT 4
SEX	Male	Male	Female	Male
AGE	Newborn, 4 weeks	Newborn	Newborn	3 years old
WEIGHT (KG)	3.5, 4.5	3.0	2.7	-
HOSPITAL	A	B	B	B

* Patient 1 underwent two procedures. -: Data not available.

**Table 2 children-09-01867-t002:** Clinical and Perioperative Characteristics.

Patient	Hospital	Age (yrs)/Wt(kg)	Procedure	Complications	Conversion to Open	Operative Time (min)	Hospital Stay (days)	Surgeon
1	A	Newborn/3.5	Type C TEF	Lack of visualization -> conversion	Yes	90	53	Visiting pediatric surgeon
		4 weeks/4.5	Thoracoscopic aortopexy	No	No	45	13	Visiting pediatric surgeon, local thoracic surgeon
2	B	Newborn/3.0	Type C TEF	Contained leak	No	90	10	Visiting pediatric surgeon
3	B	Newborn/2.7	Type C TEF	No *	No	90	-	Local and visiting pediatric surgeons
4	B	3 year old/-	Right lower lobe CPAM	No	No	90	3	Local and visiting pediatric surgeons

Abbreviations: TEF, tracheoesophageal fistula with no other anomalies; CPAM, congenital pulmonary airway malformation. -: Data not available. *: Patient died for reasons not related to the procedure itself.

**Table 3 children-09-01867-t003:** The Five S Components.

	Hospital			Hospital
Staff	A	B		Stuff	A	B
Anesthesiologists ^1^				3 mm instrument set		
Interventional Radiologists ^2^		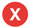	3 mm sealer	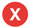	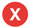
Neonatologists ^3^		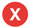	5 mm sealer	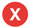	
Pediatric Intensivists			5 mm stapler *		
Pediatric Surgeons ^4^			Fluoroscopy		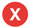
Thoracic Surgeons ^5^		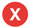	Minimally invasive tower		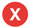
**Space**	**A**	**B**		**Systems**	**A**	**B**
OR, NICU ^6^, PICU				Formal centralized sterile processing		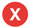
**Socialization**	**A**	**B**		Formal system for anesthesiology request	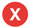	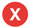
Local resident surgeons eager to assist				Patient referral ^7^		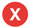
Practicing pediatric surgeons eager to learn techniques	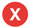			Sterilization, upkeep and storage of instruments staff ^8^	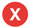	
Anesthesiologists eager to learn new procedures	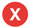			Transferring urgent cases in OR	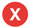	

Abbreviations: OR, operating room. * Available through outside donation. 1 Hospital B: Competent anesthesiologists with no experience in pure thoracoscopy in neonates and young children. 2 Hospital A: Full complement of diagnostic radiologic testing available with competent interpretation. 3 Hospital B: Neonatology unit staffed by pediatric residents. 4 Hospital B: Surgeons experienced and facile in advanced open thoracic procedures and have increased familiarity with minimally invasive procedures, though it was not part of their training. One adult surgeon had performed video-assisted lobectomies through a mini-thoracotomy in children. 5 Hospital A: Surgeons experienced in open thoracic procedures, but lacked experience with advanced pediatric thoracoscopy. 6 Hospital B: NICU staffed by pediatric residents. 7 Hospital B: Patients often identified and/or referred farther along the disease process. 8 Hospital B: No dedicated staff; responsibility of surgeons.

## Data Availability

Not applicable.

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
