# Peer review of "Global Health Challenges: Why the Four S’s Are Not Enough"

_children, 2022, doi:10.3390/children9121867_

Round 1

Reviewer 1 Report

This is a well written report on a personal experience introducing pediatric thoracoscopy in 2 Middle Income Countries. The authors report on a total of 4 patients who were operated on for TEF repair (3 pat.) and CPAM (1 pat.) between January 2017 and October 2018. As summarized in the conclusions, multiple barriers including the 5 "s" were identified to contribute to a sometimes seemingly unsurmountable task.

There is no doubt about that all surgeons who are priviledged to work in a HIC area, have the obligation to spread knowledge from HIC to LMIC and to support medical education, training, etc.. However, we have to notice that not everything what is feasible, is also reasonable.

Hospital A covers an area of approx. 1 million inhabitants. Were there only 2 patients presenting with TEF within 22 months? If no, what happened with all other TEF patients? According to tabel 3, hospital A provides no pediatric surgeons. How can you run a mother-child-neontaology unit without a pediatric surgeon? Vice versa, if there is no pediatric surgeon, is it the right place to treat TEF patients (who may suffer from additional congenital malformations (VACTERL))?

Many listed details (sparse experience with MIS/low numbers, limited resources, lack of neonatalogist, etc.) make the concept to introduce surgical procedures like advanced pediatric thoracoscopy debatable and questionable.

Reviewer 2 Report

***My comments are also attached as a .pdf file, for the sake of proper formatting***

-

I truly appreciated the work done by Novotny and colleagues.

Habits are hard to be changed. Challenging the status quo can get even harder, for a variety of reasons. Continued learning (as hinted in the title) is key to expand our skills, at any level. The ‘4 + 1 S’ concept sounds crucial to the successful adoption of virtually anything new, hence the results of this manuscript must be divulged.

I have a few suggestions, most of which are style/formatting edits to allow the data to speak on its own with no (or very little) effort by the reader. I am sure that production team at MDPI will provide valuable assistance with this.

1)      Table 1:

·         “Patient 4” shall be in bold, just like the previous three patients. It helps to be visually consistent across the table.

·         I would remove the final ‘dot’ from the title “PATIENT DEMOGRAPHICS.”, and capitalize only the first letters. It helps to be consistent with the titles of all other tables.

2)      Table 2:

·         I would adjust (vertical alignment) data on patient n2, 3 and 4 so that their numbers and hospital letters are uniformly aligned.

·         I would remove the words “days” next to 10 and 3 for patient 2 and 4, respectively. They are already included in the column title “Hospital Stay (days)”. Otherwise, it is just visual clutter.

·         Patient n4, under Complications: I would change “N/A” with ‘No’, to be consistent throughout the table. You truly had no complications after the lobectomy, so it is really a ‘no’ and not a ‘Not Applicable’.

·         “Time of procedure (min)”: I would change it with ‘Operative time (min)’

·         Age/Wt column: I would add a space after the backslash for patient n2, 3 and 4. It helps to be consistent with what you did for patient n1, and I believe it looks better too.

·         It strikes me a little to see that patient n3 had “No” complications, but then she died.

·   I totally understand that Table 2 is for “Clinical and Operative Characteristics”, in other words you focus specifically on patients and on the procedures performed.

·   You later describe, in the text (- - > results), the outcome indeed.

·   You hint in the table, by mean of a hyphen under ‘Hospital Stay’, that something actually happened.

·   Nevertheless, I still find it a sort of extra step for the reader to take, in imaging the patient journey.

·   So, I would try to give a very clear first image of what happened (which is the purpose of well-designed tables). This will avoid the risk to create the impression of conflicting data which, although minimal and unjustified, may arise. At least, this is what happened to me: I had a “wait a minute” moment (and I had to go back and forth a couple of times from the results/discussion back to table itself, to fully get what happened).

·   Perhaps you can edit the title of the table a bit, something like: “Clinical and Perioperative Characteristics”; or add a column at the end for the “Outcomes”; or edit the title of the column “Complications”, with something like: “Complications related to the procedure”; or add an asterisk and write: “*patient died for reasons not related to the procedure itself”.

3)      Table 3:

·         For the Staff and Stuff sub-tables: please, do position better all Ticks & Crosses (they currently are disaligned). Also, I would make the symbols a bit smaller to allow at least a bit of free-space among each row.

·         For all five sub-tables:

·   Remove “Components”: you already have it written in the title “The 5-S”. Components”. Try to avoid repetitions, unless strictly necessary.

·   Write “Hospital” just once, and subdivide the columns simply with “A” and “B”. Again, it helps to allow the data to jump out of the paper (and not to get lost in the middle of repetitions).

·         For the Staff sub-table: “Interventional Radiologists” is currently aligned to the left and justified, which I do not find easy to read: adjust the width of the table accordingly to let it be on one line, and align everything in the center (to be consistent with the other 4 sub-tables).

·         For the Socializing sub-table: Hospital A has a cross for “Practicing pediatric surgeons eager to learn techniques”. I think it is a little unfair to send across this message, as Hospital A by definition has no pediatric surgeons at all (as shown also in the Staff sub-table). If you want to keep the cross, I would recommend adding an asterisk to specify this, something like: “Hospital A had no Pediatric surgeons on staff, as shown in sub-table ‘Staff’”. I am afraid that if you do not do something like this, it just seems that Hospital A has pediatric surgeons on staff and yet they are not interested in learning. This is a powerful message that you explain very well later, in the text, depicting for example how anesthesiologists behaved at that same hospital.

·         For the Space sub-table, under abbreviations: “5Hospital B: NICU staffed by residents”. I would add the type of resident you are talking about, therefore: “NICU staffed by pediatric residents”

4)      “Results, patient n3” and Discussion, row 204-207: I think it is a little unfair the way you pointed out that something so bad happened with the resident being in charge. It is so clear to me that this is a Staff problem, that I think we have an opportunity here to convey an additional important message. We will need to phrase things in a way that we highlight the absence of a neonatologist in a hospital where neonates were operated on AND in a way that we hold residents accountable for what they do while we also protect them from a system that perhaps assigned them to the wrong service, at the wrong time.

·         Firstly, I would add an asterisk in the Staff sub-table, perhaps next to Neonatologists or directly next to the cross for Hospital B, and say that the residents were in charge of the service, indeed. The cross conveys a ‘there were no neonatologists in Hospital B’ message, but truly someone was actually there providing the care needed: simply, they perhaps were not trained enough and/or left on their own. In the Space sub-table, next to NICU, you say this and it is totally fine but, once again, what you describe in the text is mainly a Staff problem. As such, the Staff sub-table comes first for the reader hence it is in the Staff table that you have your first chance to stick to the truth.

·         Secondly, a resident by definition is still under training. By training over and over again, certain things get easier and easier to be recognized hence a “delayed recognition” (as you point out) becomes less likely to occur. A resident is a complementary resource to a department. I think we need to stick to the truth while removing the so direct link between the patient’s death and the residents being in charge, as it appears in your current writing. I am not saying you need to twist reality, at all. But certain words like ‘aggressively bagged’, ‘iatrogenic injury’, ‘over exuberant begging’ and ‘delayed recognition’ sound unfair to me. They actually may even trigger a few questions: how much time did it take for the surgical team to arrive? Are pediatric residents trained to assist neonatal patients? Where were the pediatric intensivists when it all happened? While you might need a surgeon to insert a chest tube, an intensivist would be very quick to recognize the possibility of a pneumothorax clinically. Lastly, it is very important indeed to learn from our mistakes (it is the basis of M&M conferences which have a formative and not a punitive intent). It is what makes us better people and professionals. It is very good that, after all, you now have a neonatologist in charge of the NICU.

5)      Results:

·         Row 156-157: I would add “pediatric” as follows: “…at the arrival of the pediatric surgical team to the NICU…”

·         Row 165-166: remove “Third bullet”, which was left here by mistake I guess.

6)      Discussion:

·         I would make each ‘S’ visually better: either bold them, underline them, or whatever necessary to make each dedicated paragraph easier to see and follow. At the moment, for example, Staff at row 176 is a bit lost in the middle of the rest of the text, while it is meant to be the beginning of an important paragraph and therefore it shall draw the attention there to guide the reader.

·         Row 241: “on the other hand” here seems not fitting well in the context. There is not a logical ‘on one hand’ previously, as you talk about something different anyway in the previous paragraph.

·         Row 267: you used a hyphen. Did you mean to use a colon, instead?

7)      References: if possible, add a few more. Perhaps these can be of help?

·         Pulvirenti R. et al. Pediatric Surgery and Anesthesia in Low-Middle Income Countries: Current Situation and Ethical Challenges. Front. Pediatr., 28 July 2022 Sec. Pediatric Surgery https://doi.org/10.3389/fped.2022.908699

·         Butler MW et al. Developing pediatric surgery in low- and middle-income countries: An evaluation of contemporary education and care delivery models. Seminars in Pediatric Surgery, Volume 25, Issue 1, 2016, Pages 43-50, ISSN 1055-8586, https://doi.org/10.1053/j.sempedsurg.2015.09.008.

·         Krishnaswami S. et al. The pediatric surgery workforce in low- and middle-income countries: problems and priorities. Semin Pediatr Surg 2016 Feb;25(1):32-42. doi: 10.1053/j.sempedsurg.2015.09.007. Epub 2015 Sep 21.

·         Others… … …?

Round 2

Reviewer 1 Report

Thanks to the authors to submit their revised manuscript. The manuscript experienced a significant improvement.

However, my basic concern is still the concept to present this work as "research article". As mentioned before, the concept to introduce surgical procedures like advanced pediatric thoracoscopy in such an environmentis debatable and questionable.
